# Effects of Different Feeding Durations on Ileum Length and Weight and Basal Endogenous Loss of Phosphorus in Broiler Chickens Fed a Phosphorus-Free Diet

**DOI:** 10.3390/ani13111787

**Published:** 2023-05-27

**Authors:** Su Hyun An, Changsu Kong

**Affiliations:** 1Department of Animal Science and Biotechnology, Kyungpook National University, Sangju 37224, Republic of Korea; woobi89@gmail.com; 2Department of Animal Science, Kyungpook National University, Sangju 37224, Republic of Korea; 3Research Institute of Horse Industry, Kyungpook National University, Sangju 37224, Republic of Korea

**Keywords:** broiler, endogenous loss, feeding duration, phosphorus, phosphorus free diet

## Abstract

**Simple Summary:**

The inevitable basal endogenous loss of nutrients from the body is closely associated with the animal’s metabolic functions. A purified diet formulated with highly refined feed ingredients lacking phosphorus is commonly used to determine the basal endogenous loss of phosphorus. However, such diets may alter metabolic functions, thereby affecting the basal endogenous loss of phosphorus. Moreover, feeding a purified diet for a long period of time may cause abnormal gut health, resulting in a decrease in empty gut weight and shorter intestinal segments. To the best of our knowledge, information on the effect of the feeding duration of a phosphorus-free diet on the basal endogenous loss of phosphorus is limited. We found that feeding broiler chickens a phosphorus-free diet for up to 4 days did not affect the amount of basal endogenous loss of phosphorus.

**Abstract:**

The study investigated the effects of feeding duration on the length and weight of the ileum, and basal endogenous loss (BEL) of phosphorus (P) on broiler chickens fed a P-free diet (PFD). A standard starter diet was fed to 384 birds for 15 days. On day 15, they were weighed and randomly allocated to one of three treatment groups in a randomized complete block design, with eight replicate cages per treatment and 16 birds per cage. The birds in each treatment group were fed the PFD for 2, 3, or 4 days. There was an increase in growth performance (*p* < 0.05) as the PFD feeding duration increased from 2 to 4 days. Although the relative dry weight of the ileum (mg/100 g of body weight) decreased linearly (*p* < 0.05) as the PFD feeding duration increased, the amount of dried ileal digesta (g/bird) was not affected. The BEL of P was 110.2, 128.2, and 133.6 mg/kg of dry matter intake in birds fed the PFD for 2, 3, and 4 days, respectively. Feeding a PFD to broiler chickens for 2, 3, or 4 days did not change P BEL.

## 1. Introduction

Phosphorus (P) is an essential element in the diet of poultry and serves as a component of organic compounds involved in the metabolism of energy, carbohydrates, amino acids (AAs), and fats. It is also related to cellular metabolism in poultry. Determining P digestibility is integral to formulating an appropriate diet for broiler chickens. Standardized ileal digestibility (SID) is more accurate for diet formulation than apparent ileal digestibility (AID) because SID is more additive in mixed diets than AID [1]. The SID of P is calculated by correcting the AID to account for the basal endogenous loss (BEL) of P occurring from bile secretion, salivary juice, gastric enzymes, and desquamated epithelial cells [2]. Different methods, including the regression method [3,4] and feeding a P-free diet (PFD), have been employed to measure the BEL of P in broilers [3,5,6]. These methods have advantages and disadvantages. When using the regression method, the type of feed ingredients in the basal diet and the extrapolation to zero intake on the regression line affect the BEL of P estimates. Furthermore, this method is more expensive than the PFD method because at least three data points are required to generate a reliable linear line, and thus, the number of treatments must be increased. Therefore, a PFD is commonly used in poultry experiments. Semi-purified diets, such as a PFD, which are formulated to meet nutrient requirements but without P, may have a negative impact on the normal physiology and pathophysiology of certain diseases in animals [7]. Furthermore, the lack of dietary P in birds can impair P absorption in the gut [8] and bone [9]. In addition, refined ingredients contain one main nutrient and little to no non-nutrient chemicals [10]. In this regard, even if a PFD was formulated to meet the requirements of nutrients including protein, AAs, vitamins, and minerals except for P, nutrients which are not considered might be deficient. In feed formulation, there are no defined dietary requirements for dispensable AAs, and it is unclear how much of the dispensable AAs should be supplemented to support normal growth. Deficiencies in some dispensable AAs might cause abnormal digestive function and threaten the integrity of the intestinal mucosa because dispensable AAs are involved in maintaining digestion and intestinal condition for the development and health of animals [11].

Similar to the BEL of nitrogen (N) and AAs, the most common method to determine the BEL of P is feeding a semi-purified diet without a PFD in poultry experiments. Previous studies have shown that the proportion of cornstarch to sucrose ratio can affect the BEL of N and AAs [12,13] due to differences in the digestive physiological state resulting from changes in microbial diversity and enzyme secretion [12,13,14]. For P, Mutucumarana and Ravindran [2] noted that the BEL of P differed depending on the assay diets containing sucrose, gelatin, or a casein-based PFD. In contrast, Anwar and Ravindran [15] observed that the diet composition in these PFDs did not affect the BEL of P. They used three different Ca and P-free diets containing sucrose, corn gluten meal, or dried egg albumen containing close to 0% P. This variation might be due to animal factors including age, growth rate, and feed intake, and the characteristics of the major ingredients comprising the PFD, such as differences in microbial turnover and enzyme activities [2]. Therefore, the BEL of P in birds may vary depending on the assay diets and their ingredient composition in a PFD. Several studies have been conducted to determine the BEL of P in pigs [16,17,18]. However, limited studies have been conducted to determine the BEL of P using a PFD in broilers. Therefore, it is difficult to fully understand the reasons for the variations in the BEL of P.

In addition, the rate of passage and retention time of endogenous P influences nutrient digestion [19]. A low passage rate of the digesta facilitates increased contact between nutrients, enzymes, and substrates, consequently leading to more comprehensively digested products that are absorbed by the mucosa. However, digesta from birds fed purified diets formulated with fine particle ingredients rapidly passes through the gastrointestinal tract compared to those based on coarse particle plant or animal-based diets [20]. Bryan et al. [21] observed that the amount and solubility of undigested protein in the distal ileum is influenced by dietary protein sources. They reported that the retention time of digesta in birds fed a nitrogen-free diet (NFD) was shorter in the jejunum and ileum compared to diets based on cereal grains. Therefore, it is expected that feeding a PFD formulated with refined ingredients with fine particles, similar to an NFD, results in a rapid passage rate of digesta in the intestines of broilers, which may affect the endogenous outflow of P in the distal ileum, owing to an increase in digestive enzymes not used in metabolic reactions and sloughed epithelial cells. Thus, the long-term feeding of a PFD may cause abnormal gut function in broilers due to P deficiency and fine particles, which also affect the BEL of P in birds compared with diets containing common plant or animal feed ingredients. For an NFD, poultry is typically fed for between 4 and 5 days [22]. This is due to concerns about the abnormal physiological functions of birds when they receive feed for an even longer duration. The feeding duration of a PFD varied among studies, with 2 days [2,15], 3 days [23], or 5 days [24,25]. However, information available on the effect of the feeding duration of a PFD on the BEL of P remains limited [26]. Therefore, we aimed to determine the effects of feeding duration on the growth performance, length and weight of the ileum, and the BEL of P in broiler chickens fed a PFD.

## 2. Materials and Methods

The experimental procedures were approved by the Institutional Animal Care and Use Committee of Kyungpook National University (approval number: KNU 2019-0124). The experiments were conducted in an environmentally controlled room at the Kyungpook National University (Sangju, Republic of Korea).

### 2.1. Diet, Animals, and Experimental Design

The ingredients and chemical composition of the PFD are presented in Table 1. The PFD, based on cornstarch, gelatin (Gel strength: 250 bloom, Geltech, Busan, Republic of Korea), and sucrose, was formulated to estimate the BEL of P and to meet or exceed the recommended nutrient composition [27] except for P. The PFD also contained 0.5% chromic oxide (Cr_2_O_3_) as an indigestible index for calculating the BEL of P. A total of 384 male broilers (Ross 308; 1 day of age) was obtained from a local hatchery (Samhwa GPS Breeding Agri., Inc., Boryeong, Republic of Korea). A tag was attached to the neck of each individual bird immediately after arrival at the facility. All birds were fed a commercial starter diet (Seoul feed, Incheon, Hongsung, Republic of Korea) for 15 days following hatching. On day 15, the birds were individually weighed and randomly allocated to one of three treatment groups using a randomized complete block design, with eight replicate cages per treatment group and 16 birds per cage. The birds in each treatment group were fed the PFD for 2, 3, or 4 days. All birds were offered ad libitum access to water and feed for the duration of the feeding experiment.

### 2.2. Sample Collection and Chemical Analysis

After each feeding period (2, 3, and 4 days), the corresponding group of birds was euthanized with CO_2_ and samples of the ileal digesta and ileum were collected from each bird. Growth performance, as well as the empty weight and length of the ileum, were measured on each sampling day. For each period, ileal digesta samples were collected from the distal two-thirds of the ileum, which is equivalent to the portion of the small intestine from Meckel’s diverticulum to approximately 1 cm anterior to the ileocecal junction. Representative samples of the empty ileum were then collected by removing residual fat tissues along the intestine. The collected digesta samples along with the cleaned ileum samples from the 16 birds in the same cage were pooled and stored at −20 °C for subsequent analysis. Feed, ileal digesta, and ileum samples were oven-dried at 60 °C until they attained a constant weight. Subsequently, the samples were ground and subsampled for chemical analysis. Phosphorus and calcium (Ca) concentrations in the feed, ileal digesta, and ileum were analyzed according to the procedures of the Association of Official Analytical Chemists [28]. The BEL of P was calculated using the equation described by Kong and Adeola [22]:(1)$$BEL\;of\;P\left(mg/kg\;of\;dry\;matter\;intake\left[DMI\right]\right)=\left(\frac{Cr_{diet}}{Cr_{digesta}}\right)\times P_{digesta},$$
where *Cr_diet_* (mg/kg of intake, DM basis) and *Cr_digesta_* (mg/kg of output, DM basis) are the Cr concentrations in the diet and digesta, respectively, and *P_digesta_* (mg/kg, DM basis) is the P concentration in the digesta.

### 2.3. Statistical Analysis

Data were analyzed using the MIXED procedure in SAS 9.4 [29] with a randomized complete block design. The experimental unit was the cage, the fixed variable was the PFD feeding duration, and the random variable was the initial body weight (BW) of birds, which was the block in this experiment. The growth performance, amount of ileal digesta, weight and length of the ileum, and BEL of P were analyzed for the linear and quadratic effects of different PFD feeding durations using orthogonal polynomial contrasts. Statistical significance was set at an alpha level of 0.05.

## 3. Results and Discussion

Feeding semi-purified diets containing highly refined ingredients and crystalline amino acids has been shown to result in low feed intake and a slow growth rate [30]. Furthermore, lower feed intake in birds fed semi-purified diets could exacerbate these negative impacts because the normal function and development of the intestine require certain levels of nutrient intake. Under such conditions, the secretion of digestive enzymes by animals may be insufficient, and the amounts of BEL of P, which mainly relies on bile, enzyme secretions, and sloughed epithelial cells [2], could eventually be influenced by the health of the birds. Therefore, the growth rate of birds is often recorded to indirectly confirm their physiological condition when fed PFDs, as it can indicate whether they are suffering from malnutrition due to the deficiency of essential nutrients.

The growth performance of birds fed the PFD for 2–4 days is presented in Table 2. All growth measurements increased linearly (*p* < 0.05) as the duration of PFD feeding increased from 2–4 days. In contrast, previous studies have observed weight loss and low feed intake in birds fed PFDs that are based on refined ingredients [2,15,31]. Anwar and Ravindran [15] observed different effects depending on the major ingredient of the PFD. Weight loss and reduced feed intake were observed when birds were fed a semi-purified Ca and P-free diet and a corn-gluten-meal-based PFD for 2 days, whereas birds fed a dried egg-albumen-based PFD gained weight, demonstrating that the magnitude of influence varied with dietary protein contents. Dai et al. [32] reported that dietary P deficiency decreased feed intake and growth in ducks, which may be attributed to the increased energy expenditure required to obtain P from phytate to maintain their gut microbial communities and less energy spent on nutrient utilization and growth. There is a lack of studies comparing the effect of PFD feeding duration on growth performance; thus, it is difficult to directly compare growth measurements between the present and previous studies. The expected daily gain and feed intake for birds weighing 350 g, such as Ross male broilers, were reported to range from 46 g to 53 g and from 55 g to 66 g, respectively, for a period of 2 days [33]. However, this study showed that the daily gain and feed intake were no more than 5 g and 17 g, respectively. Thus, our finding that a PFD limits the growth of chicks is in line with previous studies.

The weights and lengths of the ileum of broilers fed the PFD for 2–4 days are presented in Table 3. The absolute and relative dry weight of the ileum (mg/100 g of BW) decreased linearly (*p* < 0.05) with increasing PFD feeding duration. The absolute and relative lengths of the ileum were not significantly affected by PFD feeding duration. Several studies have reported changes in the weight and length of the small intestine of birds with increasing age [34,35]. However, there are no data on the weight and length of the intestines of broilers fed a PFD. Thus, it is difficult to directly compare the data obtained from the present study with those of other studies. 

The adverse effects of a PFD on the digestive and absorptive functions of the intestine may be a natural consequence, as phosphorus is a major component of phospholipids, which are vital constituents of cell membranes. The gastrointestinal tract is covered by a single epithelial layer that has a rapid turnover rate [36]. In broilers, the structural development of the mucosa is induced by supplementation with sufficient dietary nutrients [37]. In the present study, to meet the nutrient requirements of broilers, a PFD was formulated to contain sufficient nutrients other than P, but it is difficult to include all nutrients such as dispensable AAs. In general, the optimum feed particle size for broiler diets based on maize or sorghum is recommended to be between 600–900 µm [38]. However, the main components of a PFD are powder-fine particles such as cornstarch, granulated sucrose, and gelatin, where the particle sizes are 10–30 µm [39], 450–650 µm [40], and 381 µm, respectively. It has been reported that a coarse mash diet is associated with developing inner muscular wall motility through mechanical stimulation of the gizzard [41]. Fine particle size has an impact on reducing gizzard and pancreas mass [42], shortening relative weights of the jejunum, ileum, and whole small intestine [43], and the relative length of the entire gastrointestinal tract [44]. Additionally, the lower feed intake of birds in response to feeding a PFD compared to birds fed a typical poultry diet formulated with cereal grains may restrict essential nutrients that support their normal growth. This is because the grain-based diet contained multiple nutrients and non-nutrient chemicals, while the refined ingredients composing PFD did not [37]. The characteristics of a PFD such as fine particles, in turn, lead to a less developed upper gut and affect the passage rate of digesta and nutrient absorption. Because limited information is available on the intestinal damage caused by feeding a PFD to growing broilers, further studies are needed to better understand the relationship between the morphological functions of birds fed a PFD. 

Table 4 presents the weights of dried ileal digesta samples and the BEL of P in 17–19-day-old broilers fed a PFD for 2–4 days. Insufficient amounts of ileal digesta samples may result in difficulties in conducting chemical analyses. However, in the present study, the quantity of ileal digesta samples collected from each cage of 16 birds ranged from 9.1 to 10.6 g, which was adequate for chemical analyses, including phosphorus and chromic oxide used as an indigestible index. The BEL of P was estimated to be 110, 128, and 134 mg/kg of DMI in broilers fed a PFD for 2, 3, and 4 days, respectively, and it was not dependent on PFD feeding duration. Similar BEL of P has been reported in previous studies. When 28-day-old male broilers (from day 26–28) were fed a semi-purified PFD for 2 days, the value ranged from 110 to 133 mg/kg of DMI [15] and 104 mg/kg of DMI [2], respectively. In contrast, a higher BEL of P was reported in other studies [24,25] that used 28-day-old broilers for the feeding of a PFD for 122 h (approximately 5 days) and reported 272 [24] and 446 [25] mg/kg of DMI, respectively. It is not clear what factors would have a stronger impact on these conflicting results, but we speculate that this might be attributed to the differences in assay methods including feeding duration and ingredient compositions in the experimental diet. Moreover, these methodological differences make it difficult to directly compare the BEL of P among studies. Overall, our results demonstrate that there were no differences in the BEL of P in broilers fed a PFD with various feeding durations. 

The quantitative flow of basal endogenous at the terminal ileum are stable and only associated with DM intake [45]. The method of feeding semi-purified diets is the most common because it is relatively simple compared to other methods for determining BEL and is likely to present the most consistent results. However, Ravindran [46] found variations in published estimates. Ravindran [46] reviewed 21 papers to compare the range of the published BEL of AAs, which might be due not only to animal and dietary factors but also to the preparation and processing of ileal digesta and analytical accuracy. Therefore, several potential factors can affect the BEL of P in birds, which could lead to the variability in the estimates reported. The P in the ileal digesta is composed of both unabsorbed P from the diet and endogenous P from the body. The majority of P in the ileal digesta is composed of bile, enzyme secretions, and sloughed epithelial cells [2]. The length and width of the small intestine increase as birds age [34], and the rapid development of the small intestine regarding the height and villi occurs at an early age in broilers [19]. Feeding a PFD might delay the functional development of the bird’s intestines due to the absence of P in the diet, which might lead to adverse impacts on enzyme secretion and P absorption. In the present study, a decrease in ileum weight was found with an increase in PFD feeding duration, but the BEL of P was not influenced by PFD feeding duration. It is speculated that feeding a PFD for 2–4 days may not significantly impact the BEL of P in 15-day-old broilers because the changes in small intestine components during various PFD feeding durations were not sufficient to reveal statistical differences in the BEL of P.

In the present study, feeding a PFD for 2–4 days resulted in similar ileal DM digestibility. Generally, the DM digestibility of animals represents the amount of digested DM derived from diets at a specified level of intake. It is also observed that as PFD feeding duration and the age of birds increases, feed intake increases while the ileum weights of birds decreases. The decreased ileum weight may be a limiting factor related to the surface area for nutrient absorption in the gastrointestinal tract. Additionally, the increase in feed intake by feeding a PFD is marginal (from 8.8 to 16.9 g/bird/days from d 15 to 17, respectively), and the potential feed intake of Ross 308 broilers [47] used in the present study ranged from 74 g/bird/d on d 15 to 85 g/bird/d on d 17. Therefore, small quantities of nutrient inputs from a PFD could limit the development of the intestinal wall in digestive tracts. The BEL of nutrients can be affected by factors including DM intake, digestive enzyme secretion, and gut health related to changes in the digestive physiological status [14]. The abnormal physiological status of the gut might induce intake reduction, which interferes with the normal gastric juice and digestive enzyme secretions [48]. This eventually may lead to an underestimation of BEL [49]. However, the ileal DM digestibility was not affected by PFD feeding duration, which suggests that the intestinal status was functioning normally. Therefore, feeding a PFD for up to 4 days has no negative impacts on the digestive physiological status of birds when collecting the ileal digesta samples to determine the BEL of P.

## 4. Conclusions

In conclusion, this study confirmed that the amount of ileal digesta, estimates of the BEL of P, and dry matter ileal digestibility did not differ with an increase in PFD feeding duration from 2 to 4 days in 17 to 19-day-old male broilers. Based on these results, a PFD feeding duration of 2–4 days is recommended for collecting ileal digesta samples and determining the BEL of P for broiler chickens.

## Figures and Tables

**Table 1 animals-13-01787-t001:** Ingredients and chemical composition of the phosphorus-free diet on an as-fed basis.

Item	P-Free Diet (%)
Gelatin	20.00
Cornstarch	43.60
Sucrose	20.00
Soybean oil	3.00
Limestone	2.58
Vitamin premix ^1^	0.20
Mineral premix ^2^	0.20
Salt	0.10
Chromic oxide	0.50
Cellulose	5.00
Choline chloride	0.26
Magnesium oxide	0.09
Potassium carbonate	0.30
_L_-Histidine	0.18
_L_-Isoleucine	0.53
_L_-Leucine	0.59
_L_-Lysine	0.46
_DL_-Methionine	0.37
_L_-Cysteine	0.39
_L_-Phenylalanine	0.38
_L_-Threonine	0.54
_L_-Tryptophan	0.18
_L_-Valine	0.55
Total	100.00
Calculated value (%)	
AME (kcal/kg)	3461
Crude protein	20.74
Calcium	1.00
Non-phytate P	-
Total amino acids (%)	
Arginine	1.94
Histidine	0.35
Isoleucine	0.80
Leucine	1.20
Lysine	1.10
Methionine	0.50
Cysteine	0.40
Phenylalanine	0.72
Threonine	0.80
Tryptophan	0.20
Valine	0.90
Analyzed value (%)	
Calcium	0.98
Phosphorus	0.01

AME, apparent metabolizable energy; P, phosphorus. ^1^ The following were supplied per kg of diet: vitamin A, 24,000 IU; vitamin D_3_, 8000 IU; vitamin E, 160 mg/kg; vitamin K_3_, 8 mg/kg; vitamin B_1_, 8 mg/kg; vitamin B_2_, 20 mg/kg; vitamin B_6_, 12 mg/kg; pantothenic acid, 40 mg/kg; folic acid, 4 mg/kg; niacin, 12 mg/kg. ^2^ The following were supplied per kilogram of diet: Fe, 120 mg/kg; Cu, 320 mg/kg; Zn, 200 mg/kg; Mn, 240 mg/kg; Co, 2 mg/kg; Se, 0.6 mg/kg; I, 2.5 mg/kg.

**Table 2 animals-13-01787-t002:** Effect of phosphorus-free diet feeding duration on growth performance (g/bird) of 17–19-day-old male broilers ^1^.

Item	Feeding Durations			SEM	*p*-Values	
	2 Days	3 Days	4 Days		Linear	Quadratic
Initial BW (at day 15)	349	348	349	16.9	0.944	0.856
Final BW	354	362	371	16.5	<0.01	0.967
ADG	1.2	3.6	4.8	0.37	<0.01	0.195
ADFI	8.8	14.3	16.9	0.83	<0.01	0.001
G:F (g:g)	0.14	0.25	0.28	0.039	0.016	0.312

BW, body weight; ADG, average daily gain; ADFI, average daily feed intake; G:F, gain-to-feed ratio; SEM, standard error of the mean. ^1^ Data are the means of eight cages of 16 Ross 308 male broilers.

**Table 3 animals-13-01787-t003:** Effect of phosphorus-free diet feeding duration on the relative weight of the ileum and ileal digesta and relative length of the ileum of 17–19-day-old male broilers ^1^.

Item	Feeding Periods			SEM	*p*-Values	
	2 Days	3 Days	4 Days		Linear	Quadratic
Weight						
Dried ileum weight (g/bird)	1.19	1.16	1.13	0.063	<0.01	0.717
Dried ileum weight (mg/100 g of BW/bird)	335.7	319.0	305.2	4.73	<0.01	0.769
Length						
Ileum length (cm/bird)	44.3	45.2	47.2	1.84	0.265	0.794
Ileum length (cm/100 g of BW/bird)	12.6	12.6	13.1	0.91	0.602	0.732

BW, body weight; SEM, standard error of the mean. ^1^ Data are the means of eight cages of 16 male Ross 308 broilers.

**Table 4 animals-13-01787-t004:** Effect of phosphorus-free diet feeding duration on the weight of ileal digesta samples and basal endogenous loss of phosphorus in 17–19-day-old male broilers ^1^.

Item	Feeding Durations			SEM	*p*-Values	
	2 Days	3 Days	4 Days		Linear	Quadratic
Weight of dried ileal digesta (g/16 birds)	9.2	10.6	9.1	0.72	0.895	0.068
Weight of dried ileal digesta (g/100 g of BW/bird)	0.168	0.185	0.156	0.014	0.415	0.095
Basal endogenous loss of P (mg/kg of DMI)	110	128	134	12.8	0.212	0.691
Ileal dry matter digestibility (%)	81.3	80.2	81.2	0.85	0.959	0.374

SEM, standard error of the mean; P, phosphorus; DMI, dry matter intake. ^1^ Data are the means of eight cages of 16 male Ross 308 broilers.

## Data Availability

Raw data are held by the author and may be available upon request.

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
