# Peer review of "Effects of Different Feeding Durations on Ileum Length and Weight and Basal Endogenous Loss of Phosphorus in Broiler Chickens Fed a Phosphorus-Free Diet"

_animals, 2023, doi:10.3390/ani13111787_

Round 1
Reviewer 1 Report
General comment: the objective of the experiment was to determine the impact of feeding length on basal endogenous P loss in broilers receiving P-free diet. the concept is good and worthy of investigation. I have a few comments below which the authors will do well to consider and provide explanations for.
Specific comments
L26-7: the description of this result is vague. The author should simply state whether weight gain, FCR increased or decreased…
L55-64: I don’t understand why the discussion jumped to dispensable AA. PFD don’t have to have reduced dispensable AA, because PFD is not the same as a nitrogen-free diet. The authors may need to clarify why they connect the two factors.
L106: the authors need to define their “long-term feeding” because this is totally subjective
L129-30: why did the authors use 2, 3, or 4 days as their feeding length? Please give justification in the manuscript.
L139: two factors are intertwined here. One, the birds were not euthanized the same day, so they are not the same age when euthanized, i.e. those euthanized after 2 days of feeding would be 17 days old, and those euthanized after 4 days of feeding would be 19 days, old. Did the authors take those factors into consideration? What effect could this have?
L140: please define “ileum” in the manuscript
L168-9: this weight gain data are a bit strange to interpret. The birds were not of the same age when they were euthanized, and generally, daily rate of gain increases with age (at least during the first 4 weeks or so of age), and hence this effect might be largely due to age rather than the effect of the treatments. It doesn’t make much sense that reducing the amount of P in the diet for longer will increase weight gain.
L186-9: a more robust design would have been to have a control treatment in which the birds were not fed a P-free diet and compare the variance (difference) in daily gain between the control group and the three different treatments. Without relating the weight gain experienced here with age differences, it would be difficult to explain the effects seen.
L221-2: would it not make sense that something that impacts intestinal development would affect growth performance? You can not stimulate growth if the development of the intestinal tissue is negatively impacted. The authors should develop this anomaly in their observations, which I believe was due to age differences among the birds sampled rather than treatment differences.
L231: what does it mean the diets were analyzed for index? Perhaps the digestibility marker, chromic oxide?
L238-44: what about if the digesta quantity were expressed per g body weight? I realized the digesta was pooled for 16 birds, but the weight of the 16 birds could be taken, and the weight of the ileal digesta could be expressed as such.
L227-44: another thing to consider is whether the “dried” digesta in the different studies were dried to the same DM content. If they were not at the same DM content, the comparison could be skewed. I realize the difficulty of knowing the DM of the digesta used in different studies, but the authors could look into that possibility.
L267-8: again, don’t forget the age factor
L265-82: It might be good for the authors to explain why they chose a 2 to 4 days feeding length. Is this what is usually done? Could the feeding be extended to 7 days? Do some studies use 7 days or longer?
L289-91: this is like trying to fit a square into a circle. I think it is fair to say that this study's effect on growth performance differs from most observations, and the authors should rather concentrate on trying to answer why.
Reviewer 2 Report
The manuscript investigates the effects of feeding duration of a phosphorus-free diet on the growth performance, length and weight of the ileum, and basal endogenous loss of phosphorus in broiler chickens. The challenge of measuring basal endogenous loss is important to determine P requirements on diets. The findings are interesting and can be found in the topic of the journal, but there is a need for some major improvements in the manuscript to be accepted.
The introduction is too long and there is so much different information that doesn't fit the hypothesis and objectives of the study. For example; paragraphs 2 and 3 don't fit properly in the introduction, they can be included in the discussion. The hypothesis is not clear. Out of the 56 references, 30 are presented in the introduction, is it necessary?
The design of the experiment is an RCBD, but why did you even use a block in the experiment? What was the block? How the block is affecting your results? Why were the birds identified individually (line 125)? What was the commercial starter diet reference?)?
How can you compare the growth performance of birds at different ages (d 17,18,19?)? Is this not predictable that birds will be heavier as they grow? Why did you do a quadratic effect? Can the treatments explain a quadratic effect? Why did your results differ in growth performance from the previous studies (Anwar, M.N.; Ravindran, V., 2020; Dai, et al, 2018; (lines 173 to 189))?
The G:F does not tally with your ADG and ADFI. Please check your values and calculation.
If you had a Control treatment, would it give you a better explanation of endpoint for your results?
Minor Review:
- Line 18 - 20: Please clarify this sentence – “did not affect” “did affect”
- Line 31 – 32: Please clarify this sentence.
- Line 55 – 65: Please justify why are you talking about dispensable amino acids here?
- Line 66 – 76: Not clear what this paragraph is talking about.
- Line 82: The word “varied” does not fit in the context.
- Line 187: What do you mean by “the growth was highly restricted.”
- Line 217 – 219: What are you comparing here? Would be great to show some numbers for the readers. Maybe present some percentages.
Reviewer 3 Report
Please see the attached file.

Please see the attached file
Round 2
Reviewer 2 Report
The responses were well-addressed.